# ACTIVE LEARNING WITH PARTIAL FEEDBACK

**Peiyun Hu**[1,*] **Zachary C. Lipton**[1,3], **Anima Anandkumar**[2,3], **Deva Ramanan**[1]
[1]Carnegie Mellon University
[2]California Institute of Technology
[3]Amazon AI
peiyunh@cs.cmu.edu, zlipton@cmu.edu, anima@caltech.edu, deva@cs.cmu.edu

## ABSTRACT

While many active learning papers assume that the learner can simply ask for a label and receive it, real annotation often presents a mismatch between the form of a label (say, one among many classes), and the form of an annotation (typically *yes/no* binary feedback). To annotate examples corpora for multiclass classification, we might need to ask multiple *yes/no* questions, exploiting a label hierarchy if one is available. To address this more realistic setting, we propose active learning with partial feedback (ALPF), where the learner must actively choose both *which example to label* and *which binary question to ask*. At each step, the learner selects an example, asking if it belongs to a chosen (possibly composite) class. Each answer eliminates some classes, leaving the learner with a *partial label*. The learner may then either ask more questions about the same example (until an exact label is uncovered) or move on immediately, leaving the first example partially labeled. Active learning with partial labels requires (i) a sampling strategy to choose (example, class) pairs, and (ii) learning from partial labels between rounds. Experiments on *Tiny ImageNet* demonstrate that our most effective method improves 26% (relative) in top-1 classification accuracy compared to i.i.d. baselines and standard active learners given 30% of the annotation budget that would be required (naively) to annotate the dataset. Moreover, ALPF-learners fully annotate TinyImageNet at 42% lower cost. Surprisingly, we observe that accounting for per-example annotation costs can alter the conventional wisdom that active learners should solicit labels for *hard* examples.

## 1 INTRODUCTION

Given a large set of unlabeled images, and a budget to collect annotations, how can we learn an accurate image classifier most economically? Active Learning (AL) seeks to increase data efficiency by strategically choosing which examples to annotate. Typically, AL treats the labeling process as atomic: every annotation costs the same and produces a correct label. However, large-scale multi-class annotation is seldom atomic; we can't simply ask a crowd-worker to select one among 1000 classes if they aren't familiar with our ontology. Instead, annotation pipelines typically solicit feedback through simpler mechanisms such as yes/no questions. For example, to construct the 1000-class ImageNet dataset, researchers first filtered candidates for each class via Google Image Search, then asking crowd-workers questions like "Is there a Burmese cat in this image?" (Deng et al., 2009). For tasks where the Google trick won't work, we might exploit class hierarchies to drill down to the *exact* label. Costs scale with the number of questions asked. Thus, real-world annotation costs can vary per example (Settles, 2011).

**We propose *Active Learning with Partial Feedback* (ALPF)**, asking, *can we cut costs by actively choosing both which examples to annotate, and which questions to ask?* Say that for a new image, our current classifier

---

*This work was done while the author was an intern at Amazon AI.

places 99% of the predicted probability mass on various dog breeds. Why start at the top of the tree – *"is this an artificial object?"* – when we can cut costs by jumping straight to dog breeds (Figure 1)?

ALPF proceeds as follows: In addition to the class labels, the learner possesses a *pre-defined* collection of *composite classes*, e.g. *dog* ⊃ *bulldog*, *mastiff*, .... At each round, the learner selects an (example, class) pair. The annotator responds with binary feedback, leaving the learner with a *partial label*. If only the *atomic* class label remains, the learner has obtained an *exact label*. For simplicity, we focus on hierarchically-organized collections—trees with atomic classes as leaves and composite classes as internal nodes.

For this to work, we need a hierarchy of concepts *familiar* to the annotator. Imagine asking an annotator *"is this a **foo**?"* where *foo* represents a category comprised of 500 *random* ImageNet classes. Determining class membership would be onerous for the same reason that providing an exact label is: It requires the annotator be familiar with an enormous list of seemingly-unrelated options before answering. On the other hand, answering *"is this an animal?"* is easy despite *animal* being an extremely coarse-grained category —because most people already know what an animal is.

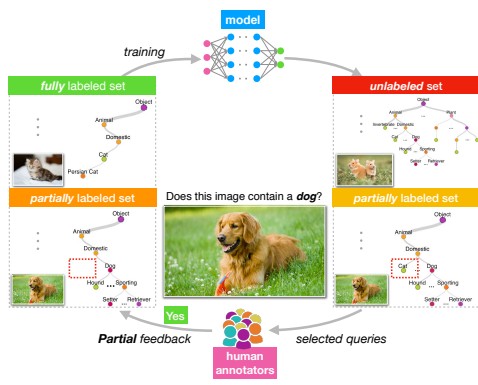

Figure 1: Workflow for an ALPF learner.

We use *active questions* in a few ways. To start, in the simplest setup, we can select samples at random but then once each sample is selected, choose questions actively until finding the label:

| | |
|---|---|
| ML: "Is it a dog?" | Human: Yes! |
| ML: "Is it a poodle?" | Human: No! |
| ML: "Is it a hound?" | Human: Yes! |
| ML: "Is it a Rhodesian ?" | Human: No! |
| **ML: "Is it a Dachsund?"** | **Human: Yes!** |

In ALPF, we go one step further. Since our goal is to produce accurate classifiers on tight budget, should we necessarily label each example to completion? After each question, ALPF learners have the option of choosing a *different example* for the next binary query. Efficient learning under ALPF requires (i) good strategies for choosing (example, class) pairs, and (ii) techniques for learning from the partially-labeled data that results when labeling examples to completion isn't required.

We first demonstrate an effective scheme for learning from partial labels. The predictive distribution is parameterized by a softmax over all classes. On a per-example basis, we convert the multiclass problem to a binary classification problem, where the two classes correspond to the subsets of *potential* and *eliminated* classes. We determine the total probability assigned to *potential* classes by summing over their softmax probabilities. For active learning with partial feedback, we introduce several acquisition functions for soliciting partial labels, selecting questions among all (example, class) pairs. One natural method, expected information gain (EIG) generalizes the classic maximum entropy heuristic to the ALPF setting. Our two other heuristics, EDC and ERC, select based on the number of labels that we *expect* to see *eliminated from* and *remaining in* a given partial label, respectively.

We evaluate ALPF learners on CIFAR10, CIFAR100, and Tiny ImageNet datasets. In all cases, we use WordNet to impose a hierarchy on our labels. Each of our experiments simulates rounds of active learning, starting with a small amount of i.i.d. data to warmstart the models, and proceeding until all examples are

exactly labeled. We compare models by their test-set accuracy after various amounts of annotation. Experiments show that ERC sampling performs best. On *TinyImageNet*, with a budget of 250k binary questions, ALPF improves in accuracy by 26% (relative) and 8.1% (absolute) over the i.i.d. baseline. Additionally, ERC & EDC fully annotate the dataset with just $491k$ and $484k$ examples binary questions, respectively (vs 827k), a 42% reduction in annotation cost. Surprisingly, we observe that taking disparate annotation costs into account may alter the conventional wisdom that active learners should solicit labels for *hard* examples. In ALPF, *easy* examples might yield less information, but are cheaper to annotate.

## 2 ACTIVE LEARNING WITH PARTIAL FEEDBACK

By $x \in \mathcal{R}^d$ and $y \in \mathcal{Y}$ for $\mathcal{Y} = \{\{1\}, ..., \{k\}\}$, we denote feature vectors and labels. Here $d$ is the feature dimension and $k$ is the number of *atomic* classes. By *atomic* class, we mean that they are indivisible. As in conventional AL, the agent starts off with an unlabeled training set $\mathcal{D} = \{x_1, ..., x_n\}$.

**Composite classes** We also consider a pre-specified collection of composite classes $\mathcal{C} = \{c_1, ..., c_m\}$, where each composite class $c_i \subset \{1, ..., k\}$ is a subset of labels such that $|c_i| \geq 1$. Note that $\mathcal{C}$ includes both the atomic and composite classes. In this paper's empirical section, we generate composite classes by imposing an existing lexical hierarchy on the class labels (Miller, 1995).

**Partial labels** For an example $i$, we use *partial label* to describe any element $\tilde{y}_i \subset \{1, ..., k\}$ such that $\tilde{y}_i \supset y_i$. We call $\tilde{y}_i$ a *partial label* because it may rule out some classes, but doesn't fully indicate underlying atomic class. For example, *dog* = {*akita, beagle, bulldog, ...*} is a valid partial label when the true label is {*bulldog*}. An ALPF learner eliminates classes, obtaining successively smaller partial labels, until only one (the *exact label*) remains. To simplify notation, in this paper, by an example's *partial label*, we refer to the smallest partial label available based on the already-eliminated classes. At any step $t$ and for any example $i$, we use $\tilde{y}_i^{(t)}$ to denote the current partial label. The initial partial label for every example is $\tilde{y}^0 = \{1, ..., k\}$ An *exact label* is achieved when the partial label $\tilde{y}_i = y_i$.

**Partial Feedback** The set of possible questions $\mathcal{Q} = \mathcal{X} \times \mathcal{C}$ includes all pairs of examples and composite classes. An ALPF learner interacts with annotators by choosing questions $q \in \mathcal{Q}$. Informally, we pick a question $q = (x_i, c_j)$ and ask the annotator, *does $x_i$ contain a $c_j$?* If the queried example's label belongs to the queried composite class ($y_i \subset c_j$), the answer is 1, else 0.

Let $\alpha_q$ denote the binary answer to question $q \in \mathcal{Q}$. Based on the partial feedback, we can compute the new partial label $\tilde{y}^{(t+1)}$ according to Eq. equation 1,

$$\tilde{y}^{(t+1)} = \begin{cases} \tilde{y}^{(t)} \setminus c & \text{if } \alpha = 0 \\ \tilde{y}^{(t)} \setminus \bar{c} & \text{if } \alpha = 1 \end{cases} \tag{1}$$

Note that here $\tilde{y}^{(t)}$ and $c$ are sets, $\alpha$ is a bit, $\bar{c}$ is a set complement, and that $\tilde{y}^{(t)} \setminus \bar{c}$ and $\tilde{y}^{(t)} \setminus c$ are set subtractions to eliminate classes from the partial label based on the answer.

**Learning Process** The learning process is simple: At each round $t$, the learner selects a pair $(x, c)$ for labeling. Note that a rational agent will never select either (i) an example for which the exact label is known, or (ii) a pair $(x, c)$ for which the answer is already known, e.g., if $c \supset \tilde{y}^{(t)}$ or $c \cap \tilde{y}^{(t)} = \emptyset$. After receiving binary feedback, the agent updates the corresponding partial label $\tilde{y}^{(t)} \to \tilde{y}^{(t+1)}$, using Equation 1. The agent then re-estimates its model, using all available non-trivial partial labels and selects another question $q$. In batch-mode, the ALPF learner re-estimates its model once per $T$ queries which is necessary when training is expensive (e.g. deep learning). We summarize the workflow of a ALPF learner in Algorithm 1.

**Objectives** We state two goals for ALPF learners. First, we want to learn predictors with low error (on exactly labeled i.i.d. holdout data), given a fixed annotation budget. Second, we want to fully annotate datasets at the lowest cost. In our experiments (Section 3), a ALPF strategy dominates on both tasks.

**Algorithm 1** Active Learning with Partial Feedback

---

**Input:** $\mathbf{X} \leftarrow (\mathbf{x}_1, \ldots, \mathbf{x}_N), \mathbf{Q} \leftarrow (\mathbf{q}_1, \ldots, \mathbf{q}_M),$ $K, T.$
**Input:** $\mathcal{D} \leftarrow [\boldsymbol{x}_i]_{i=1}^N, \mathcal{C} \leftarrow [c_j]_{j=1}^M, k, T$
**Initialize:** $\tilde{y}_i^{(0)} \leftarrow \{1, \ldots, k\}, \theta \leftarrow \theta^{(0)}, t \leftarrow 0$
**repeat**
   Score every $(\boldsymbol{x}_i, c_j)$ with $\theta$
   **repeat**
      Select $(\boldsymbol{x}_{i^*}, c_{j^*})$ with the best score
      Query $c_{j^*}$ on data $\boldsymbol{x}_{i^*}$
      Receive feedback $\alpha$
      Update $\tilde{y}_{i^*}^{(t+1)}$ according to $\alpha$
      $t \leftarrow t + 1$
   **until** $(t \bmod T = 0)$ or $(\forall i, |\tilde{y}_i^{(t)}| = 1)$
   $\theta \leftarrow \arg\min_\theta \mathcal{L}(\theta)$
**until** $\forall i, |\tilde{y}_i^{(t)}| = 1$ or $t$ exhausts budget

---

Table 1: Learning from partial labels on Tiny ImageNet. These results demonstrate the usefulness of our training scheme absent the additional complications due to ALPF. In each row, $\gamma\%$ of examples are assigned labels at the *atomic class* (Level 0). Levels 1, 2, and 4 denote progressively coarser composite labels tracing through the WordNet hierarchy.

| $\gamma(\%)$ | $\gamma$ | $(1-\gamma)$ | | |
| --- | --- | --- | --- | --- |
| | Level 0 | Level 1 | Level 2 | Level 4 |
| 20 | 0.285 | **+0.113** | +0.086 | +0.025 |
| 40 | 0.351 | +0.079 | +0.056 | +0.016 |
| 60 | 0.391 | +0.051 | +0.036 | +0.018 |
| 80 | 0.432 | +0.015 | +0.017 | -0.009 |
| 100 | 0.441 | - | - | - |

## 2.1 LEARNING FROM PARTIAL LABELS

We now address the task of learning a multiclass classifier from partial labels, a fundamental requirement of ALPF, regardless of the choice of sampling strategy. At time $t$, our model $\hat{y}(y, \boldsymbol{x}, \theta^{(t)})$ parameterised by parameters $\theta^{(t)}$ estimates the conditional probability of an atomic class $y$. For simplicity, when the context is clear, we will use $\hat{\boldsymbol{y}}$ to designate the full vector of predicted probabilities over all classes. The probability assigned to a partial label $\tilde{y}$ can be expressed by marginalizing over the atomic classes that it contains: $\hat{p}(\tilde{y}^{(t)}, \boldsymbol{x}, \theta^{(t)}) = \sum_{y \in \tilde{y}^{(t)}} \hat{y}(y, \boldsymbol{x}, \theta^{(t)})$. We optimize our model by minimizing the log loss:

$$\mathcal{L}(\theta^{(t)}) = -\frac{1}{n} \sum_{i=1}^n \log\left[\hat{p}(\tilde{y}_i^{(t)}, \boldsymbol{x}_i, \theta^{(t)})\right] \tag{2}$$

Note that when every example is exactly labeled, our loss function simplifies to the standard cross entropy loss often used for multi-class classification. Also note that when every partial label contains the full set of classes, all partial labels have probability 1 and the update is a no-op. Finally, if the partial label indicates a composite class such as *dog*, and the predictive probability mass is exclusively allocated among various breeds of dog, our loss will be 0. Models are only updated when their predictions disagree (to some degree) with the current partial label.

## 2.2 SAMPLING STRATEGIES

**Expected Information Gain (EIG):** Per classic uncertainty sampling, we can quantify a classifer's uncertainty via the entropy of the predictive distribution. In AL, each query returns an exact label, and thus the post-query entropy is always 0. In our case, each answer to the query yields a different partial label. We use the notation $\hat{\boldsymbol{y}}_0$, and $\hat{\boldsymbol{y}}_1$ to denote consequent predictive distributions for each answer (no or yes). We generalize maximum entropy to ALPF by selecting questions with greatest *expected reduction in entropy*.

$$EIG_{(\boldsymbol{x},c)} = S(\hat{\boldsymbol{y}}) - [\hat{p}(c, \boldsymbol{x}, \theta)S(\hat{\boldsymbol{y}}_1) + (1 - \hat{p}(c, \boldsymbol{x}, \theta))S(\hat{\boldsymbol{y}}_0)] \tag{3}$$

where $S(\cdot)$ is the entropy function. It's easy to prove that EIG is maximized when $\hat{p}(c, \boldsymbol{x}, \theta) = 0.5$.

**Expected Remaining Classes (ERC):** Next, we propose ERC, a heuristic that suggests arriving as quickly as possible at exactly-labeled examples. At each round, ERC selects those examples for which the expected number of remaining classes is fewest:

$$ERC_{(\boldsymbol{x},c)} = \hat{p}(c,\boldsymbol{x},\theta)||\hat{\boldsymbol{y}}_1||_0 + (1 - \hat{p}(c,\boldsymbol{x},\theta))||\hat{\boldsymbol{y}}_0||_0, \tag{4}$$

where $||\hat{\boldsymbol{y}}_\alpha||$ is the size of the partial label following given answer $\alpha$. ERC is minimized when the result of the feedback will produce an exact label with probability 1. For a given example $\boldsymbol{x}_i$, if $||\hat{\boldsymbol{y}}_i||_0 = 2$ containing only the potential classes (e.g.) *dog* and *cat*, then with certainty, ERC will produce an exact label by querying the class $\{dog\}$ (or equivalently $\{cat\}$). This heuristic is inspired by Cour et al. (2011), which shows that the partial classification loss (what we optimize with partial labels) is an upper bound of the true classification loss (as if true labels are available) with a linear factor of $\frac{1}{1-\varepsilon}$, where $\varepsilon$ is ambiguity degree and $\varepsilon \propto |\tilde{y}|$. By selecting $q \in \mathcal{Q}$ that leads to the smallest $|\tilde{y}|$, we can tighten the bound to make optimization with partial labels more effective.

**Expected Decrease in Classes (EDC):** More in keeping with the traditional goal of minimizing uncertainty, we might choose EDC, the sampling strategy which we expect to result in the greatest reduction in the number of potential classes. We can express EDC as the difference between the number of potential labels (known) and the expected number of potential labels remaining: $EDC_{(\boldsymbol{x},c)} = |\tilde{y}^{(t)}| - ERC_{(\boldsymbol{x},c)}$.

## 3 EXPERIMENTS

We evaluate ALPF algorithms on the CIFAR10, CIFAR100, and Tiny ImageNet datasets, with training sets of 50k, 50k, and 100k examples, and 10, 100, and 200 classes respectively. After imposing the Wordnet hierarchy on the label names, the size of the set of possible binary questions $|\mathcal{C}|$ for each dataset are 27, 261, and 304, respectively. The number of binary questions between re-trainings are 5k, 15k, and 30k, respectively. By default, we warm-start each learner with the same 5% of training examples selected i.i.d. and exactly labeled. Warm-starting has proven essential in other papers combining deep and active learning (Shen et al., 2018). Our own analysis (Section 3.3) confirms the importance of warm-starting although the affect appears variable across acquisition strategies.

**Model** For each experiment, we adopt the widely-popular ResNet-18 architecture (He et al., 2016). Because we are focused on active learning and thus seek fundamental understanding of this new problem formulation, we do not complicate the picture with any fine-tuning techniques. Note that some leaderboard scores circulating on the Internet appear to have far superior numbers. This owes to pre-training on the full ImageNet dataset (from which Tiny-ImageNet was subsampled and downsampled), constituting a target leak.

We initialize weights with the *Xavier* technique (Glorot and Bengio, 2010) and minimize our loss using the Adam (Kingma and Ba, 2014) optimizer, finding that it outperforms SGD significantly when learning from partial labels. We use the same learning rate of $0.001$ for all experiments, first-order momentum decay ($\beta_1$) of $0.9$, and second-order momentum decay ($\beta_2$) of $0.999$. Finally, we train with mini-batches of 200 examples and perform standard data augmentation techniques including random cropping, resizing, and mirror-flipping. We implement all models in MXNet and have posted our code publicly[1].

**Re-training** Ideally, we might update models after each query, but this is too costly. Instead, following Shen et al. (2018) and others, we alternately query labels and update our models in rounds. We warm-start all experiments with 5% labeled data and iterate until every example is exactly labeled. At each round, we re-train our classifier from scratch with random initialization. While we could initialize the new classifier with the previous best one (as in Shen et al. (2018)), preliminary experiments showed that this faster convergence comes at the cost of worse performance, perhaps owing to severe over-fitting to labels acquired early in training. In all experiments, for simplicity, we terminate the optimization after 75 epochs. Since

---

[1]Our implementations of ALPF learners are available at: https://github.com/peiyunh/alpf

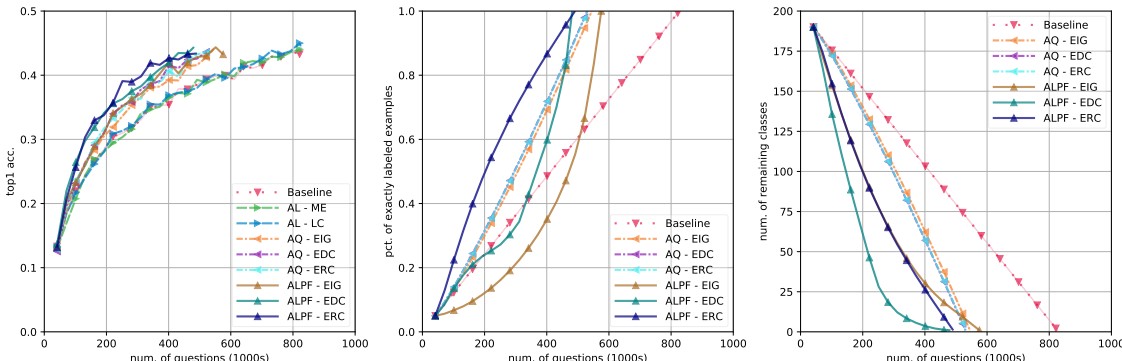

Figure 2: The progression of top1 classification accuracy (left), percentage of exactly labeled training examples (middle), and average number of remaining classes (right).

$30k$ questions per re-training (for TinyImagenet) seems infrequent, we compared against 10x more frequent re-training More frequent training conferred no benefit (Appendix B).

### 3.1 LEARNING FROM PARTIAL LABELS

Since the success of ALPF depends in part on learning from partial labels, we first demonstrate the efficacy of learning from partial labels with our loss function when the partial labels are given a priori. In these experiments we simulate a partially labeled dataset and show that the learner achieves significantly better accuracy when learning from partial labels than if it excluded the partial labels and focused only on exactly annotated examples. Using our WordNet-derived hierarchy, we conduct experiments with partial labels at different levels of granularity. Using partial labels from one level above the leaf, *German shepherd* becomes *dog*. Going up two levels, it becomes *animal*.

We first train a standard multi-class classifier with $\gamma$ (%) *exactly labeled* training data and then another classifier with the remaining $(1 - \gamma)\%$ *partially labeled* at a different granularity (level of hierarchy). We compare the classifier performance on holdout data both *with* and *without* adding *partial labels* in Table 1. We make two key observations: (i) additional coarse-grained partial labels improve model accuracy (ii) as expected, the improvement diminishes as partial label gets coarser. These observations suggest we can learn effectively given a mix of exact and partial labels.

### 3.2 SAMPLING STRATEGIES

**Baseline** This learner samples examples at random. Once an example is sampled, the learner applies top-down binary splitting—choosing the question that most evenly splits the probability mass, see Related Work for details— with a uniform prior over the classes until that example is exactly labeled.

**AL** To disentangle the effect of active sampling of questions and samples, we compare to conventional AL approaches selecting examples with uncertainty sampling but selecting questions as baseline.

**AQ** *Active questions* learners, choose examples at random but use partial feedback strategies to efficiently label those examples, moving on to the next example after finding an example's exact label.

**ALPF** ALPF learners are free to choose any (example, question) pair at each turn, Thus, unlike AL and AQ, ALPF learners commonly encounter partial labels during training.

**Results** We run all experiments until fully annotating the training set. We then evaluate each method from two perspectives: classification and annotation. We measure each classifiers' top-1 accuracy at each annotation budget. To quantify annotation performance, we count the number questions required to *exactly label* all training examples. We compile our results in Table 2, rounding costs to 10%, 20% etc. The budget includes the (5%) i.i.d. data for warm-starting. Some key results: (i) vanilla active learning does not improve over i.i.d. baselines, confirming similar observations on image classification by Sener and Savarese (2017); (ii) AQ provides a dramatic improvement over baseline. The advantage persists throughout training. These learners sample examples randomly and label to completion (until an exact label is produced) before moving on, differing only in how efficiently they annotate data. (iii) On Tiny ImageNet, at 30% of budget, ALPF-ERC outperforms AQ methods by 4.5% and outperforms the i.i.d. baseline by 8.1%.

## 3.3 Diagnostic analyses

First, we study how different amounts of warm-starting affects ALPF learners' performance with a small set of i.i.d. labels. Second, we compare the selections due to ERC and EDC to those produced through uncertainty sampling. Third, we note that while EDC and ERC appear to perform best on our problems, they may be vulnerable to excessively focusing on classes that are trivial to recognize. We examine this setting via an adversarial dataset intended to break the heuristics.

**Warm-starting** We compare the performance of each strategy under different percentages (0%, 5%, and 10%) of pre-labeled i.i.d. data (Figure 5, Appendix A). Results show that ERC works properly even without warm-starting, while EIG benefits from a 5% warm-start and EDC suffers badly without warm-starting. We observe that 10% warm-starting yields no further improvement.

**Sample uncertainty** Classic uncertainty sampling chooses data of high uncertainty. This question is worth re-examining in the context of ALPF. To analyze the behavior of ALPF learners vis-a-vis uncertainty we plot average prediction entropy of sampled data for ALPF learners with different sampling strategies (Figure 3). Note that ALPF learners using EIG pick high-entropy data, while ALPF learners with EDC and ERC choose examples with lower entropy predictions. The (perhaps) surprising performance of EDC and ERC may owe to the cost structure of ALPF. While labels for examples with low-entropy predictions confer less information, they also come at lower cost.

**Adversarial setting** Because ERC goes after "easy" examples, we test its behavior on a simulated dataset where 2 of the *CIFAR10* classes (randomly chosen) are trivially easy. We set all pixels white for one class all pixels black for the other. We plot the label distribution among the selected data over rounds of selection in against that on the unperturbed *CIFAR10* in Figure 4. As we can see, in the normal case, EIG splits its budget among all classes roughly evenly while EDC and ERC focus more on different classes at different stages. In the adversarial case, EIG quickly learns the easy classes, thereafter focusing on the others until they are exhausted, while EDC and ERC concentrate on exhausting the easy ones first. Although EDC and ERC still manage to label all data with less total cost than EIG, this behavior might cost us when we have trivial classes, especially when our unlabeled dataset is enormous relative to our budget.

# 4 Related work

**Binary identification:** Efficiently finding answers with yes/no questions is a classic problem (Garey, 1972) dubbed *binary identification*. Hyafil and Rivest (1976) proved that finding the optimal strategy given an arbitrary set of binary tests is NP-complete. A well-known greedy algorithm called *binary splitting* (Garey and Graham, 1974; Loveland, 1985), chooses questions that most evenly split the probability mass.

**Active learning:** Our work builds upon the AL framework (Box and Draper, 1987; Cohn et al., 1996; Settles, 2010) (vs. i.i.d labeling). Classical AL methods select examples for which the current predictor is

Table 2: Results on Tiny ImageNet (N/A indicates data has been fully labeled)

| | Annotation Budget (w.r.t. baseline labeling cost) | | | | | | Labeling Cost |
|---|---|---|---|---|---|---|---|
| | 10% | 20% | 30% | 40% | 50% | 100% | |
| **TinyImageNet** | | | | | | | |
| Baseline | 0.186 | 0.266 | 0.310 | 0.351 | 0.354 | 0.441 | 827k |
| AL - ME | 0.169 | 0.269 | 0.303 | 0.347 | 0.365 | - | 827k |
| AL - LC | 0.184 | 0.262 | 0.313 | 0.355 | 0.369 | - | 827k |
| AQ - EIG | 0.186 | 0.283 | 0.336 | 0.381 | 0.393 | - | 545k |
| AQ - EDC | 0.196 | 0.291 | 0.353 | 0.386 | 0.415 | - | 530k |
| AQ - ERC | 0.194 | 0.295 | 0.346 | 0.394 | 0.406 | - | 531k |
| ALPF - EIG | 0.203 | 0.289 | 0.351 | 0.384 | 0.420 | - | 575k |
| ALPF - EDC | **0.220** | 0.319 | 0.363 | 0.397 | 0.420 | - | 482k |
| ALPF - ERC | 0.207 | **0.330** | **0.391** | **0.419** | **0.427** | - | 491k |
| **CIFAR100** | | | | | | | |
| Baseline | 0.252 | 0.340 | 0.412 | 0.437 | 0.469 | 0.537 | 337k |
| AL - ME | 0.237 | 0.321 | 0.388 | 0.419 | 0.458 | - | 337k |
| AL - LC | 0.247 | 0.332 | 0.398 | 0.432 | 0.468 | - | 337k |
| AQ - EIG | 0.266 | 0.354 | 0.443 | 0.485 | 0.502 | - | 208k |
| AQ - EDC | 0.264 | 0.366 | 0.439 | 0.483 | 0.508 | - | 215k |
| AQ - ERC | 0.256 | 0.366 | 0.453 | 0.479 | 0.496 | - | 215k |
| ALPF - EIG | 0.263 | 0.341 | 0.423 | 0.466 | 0.497 | - | 235k |
| ALPF - EDC | **0.281** | 0.367 | 0.442 | 0.479 | 0.518 | - | 193k |
| ALPF - ERC | 0.273 | **0.379** | **0.464** | **0.502** | **0.526** | - | 187k |
| **CIFAR10** | | | | | | | |
| Baseline | 0.645 | 0.718 | 0.757 | 0.778 | 0.792 | 0.829 | 170k |
| AL - ME | 0.663 | 0.709 | 0.759 | 0.763 | 0.800 | - | 170k |
| AL - LC | 0.644 | 0.724 | 0.753 | 0.780 | 0.792 | - | 170k |
| AQ - EIG | 0.654 | 0.747 | 0.791 | 0.806 | 0.823 | - | 89k |
| AQ - EDC | 0.675 | 0.746 | 0.784 | 0.789 | 0.826 | - | 95k |
| AQ - ERC | 0.682 | 0.750 | 0.771 | 0.811 | 0.822 | - | 96k |
| ALPF - EIG | 0.673 | 0.741 | 0.786 | 0.815 | 0.813 | - | 124k |
| ALPF - EDC | **0.676** | **0.752** | **0.797** | 0.832 | N/A | - | 74k |
| ALPF - ERC | 0.670 | 0.743 | **0.797** | **0.833** | N/A | - | 74k |

most uncertain, according to various notions of uncertainty: Dagan and Engelson (1995) selects examples with *maximum entropy* (ME) predictive distributions, while Culotta and McCallum (2005) uses the *least confidence* (LC) heuristic, sorting examples in ascending order by the probability assigned to the argmax. Settles et al. (2008) notes that annotation costs may vary across data points suggesting cost-aware sampling heuristics but doesn't address the setting when costs change dynamically during training as a classifier grows stronger. Luo et al. (2013) incorporates structure among outputs into an active learning scheme in the context of structured prediction. Mo et al. (2016) addresses hierarchical label structure in active learning interestingly in a setting where subclasses are *easier* to learn. Thus they query classes more fine-grained than the targets, while we solicit feedback on more *general* categories.

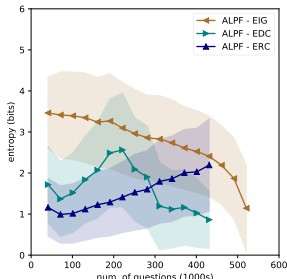 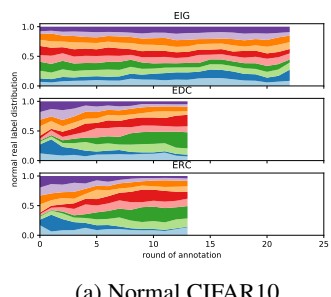 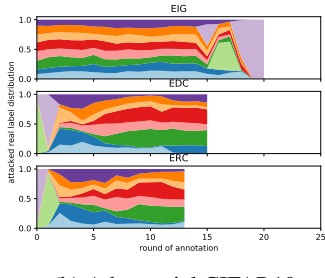

(a) Normal CIFAR10       (b) Adversarial CIFAR10

Figure 3: Classifier confidence (entropy of softmax layer) on *selected* examples.

Figure 4: Label distribution among selected examples for CIFAR 10 (left) and adversarially perturbed CIFAR 10 (right). Light green and light purple mark the two classes made artificially easy.

**Deep Active Learning** Deep Active Learning (DAL) has recently emerged as an active research area. Wang et al. (2016) explores a scheme that combines traditional heuristics with pseudo-labeling. Gal et al. (2017) notes that the softmax outputs of neural networks do not capture epistemic uncertainty (Kendall and Gal, 2017), proposing instead to use Monte Carlo samples from a dropout-regularized neural network to produce uncertainty estimates. DAL has demonstrated success on NLP tasks. Zhang et al. (2017) explores AL for sentiment classification, proposing a new sampling heuristic, choosing examples for which the expected update to the word embeddings is largest. Recently, Shen et al. (2018) matched state of the art performance on named entity recognition, using just 25% of the training data. Kampffmeyer et al. (2016) and Kendall et al. (2015) explore other measures of uncertainty over neural network predictions.

**Learning from partial labels** Many papers on learning from partial labels (Grandvalet and Bengio, 2004; Nguyen and Caruana, 2008; Cour et al., 2011) assume that partial labels are given a priori and fixed. Grandvalet and Bengio (2004) formalizes the partial labeling problem in the probabilistic framework and proposes a minimum entropy based solution. Nguyen and Caruana (2008) proposes an efficient algorithm to learn classifiers from partial labels within the max-margin framework. Cour et al. (2011) addresses desirable properties of partial labels that allow learning from them effectively. While these papers assume a fixed set of partial labels, we *actively* solicit partial feedback. This presents new algorithmic challenges: (i) the partial labels for each data point changes across training rounds; (ii) the partial labels result from active selection, which introduces bias; and (iii) our problem setup requires a sampling strategy to choose questions.

## 5   Conclusion

Our experiments validate the active learning with partial feedback framework on large-scale classification benchmarks. The best among our proposed ALPF learners fully labels the data with 42% fewer binary questions as compared to traditional active learners. Our diagnostic analysis suggests that in ALPF, it's sometimes more efficient to start with "easier" examples that can be cheaply annotated rather than with "harder" data as often suggested by traditional active learning.

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

## A  WARM-STARTING PLOT

Figure 5 compares our strategies under various amounts of warm-starting with pre-labeled i.i.d. data.

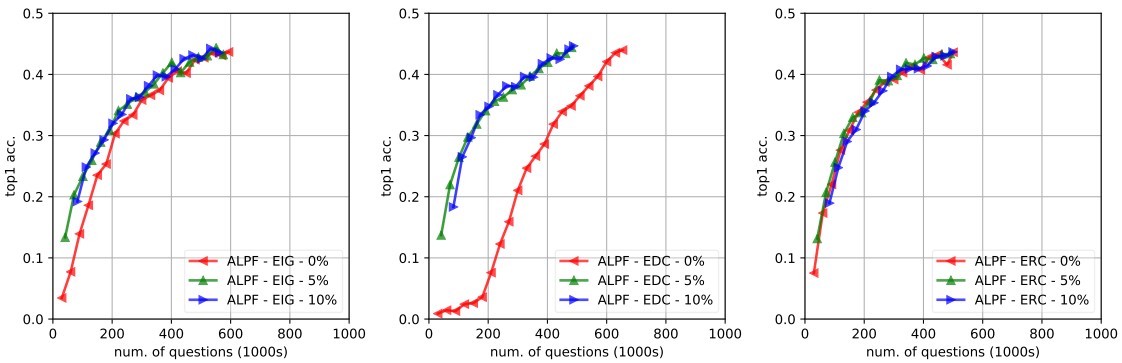

Figure 5: This plot compares our models under various amounts of warm-starting with pre-labeled i.i.d. data. We find that on the investigated datasets, ERC does benefit from warm-starting. However, absent warm-starting, EIG performs significantly worse and EDC suffers even more. We find that 5% warmstarting helps these two models and that for both, increasing warm-starting from 5% up to 10% does not lead to further improvements.

## B  UPDATING MODELS MORE FREQUENTLY

On Tiny ImageNet, we normally re-initialize and train models from scratch for 75 epochs after every 30K questions. Since we found re-initialization is crucial for good performance, to ensure a fair comparison, we keep the same re-initialization frequency (i.e. every 30K questions) while updating the model by fine-tuning 5 epochs after every 3K questions. This results in 10X faster model updating frequency. As in Figure 6 and Table 3, results show only ALPF-EDC and ALPF-ERC seem to benefit from updating 10 times more frequently

Table 3: Updating models after every 30K questions (1X) vs. after every 3K (10X)

| | Annotation Budget | | | | | | Labeling Cost |
|---|---|---|---|---|---|---|---|
| | 10% | 20% | 30% | 40% | 50% | 100% | |
| **TinyImageNet** | | | | | | | |
| Baseline | 0.186 | 0.266 | 0.310 | 0.351 | 0.354 | 0.441 | 827k |
| AL - ME | 0.169 | 0.269 | 0.303 | 0.347 | 0.365 | - | 827k |
| AL - LC | 0.184 | 0.262 | 0.313 | 0.355 | 0.369 | - | 827k |
| AL - ME - 10X | 0.177 | 0.260 | 0.304 | 0.341 | 0.359 | - | |
| AL - LC - 10X | 0.188 | 0.257 | 0.308 | 0.347 | 0.369 | - | |
| AQ - EIG | 0.186 | 0.283 | 0.336 | 0.381 | 0.393 | - | 545k |
| AQ - EDC | 0.196 | 0.291 | 0.353 | 0.386 | 0.415 | - | 530k |
| AQ - ERC | 0.194 | 0.295 | 0.346 | 0.394 | 0.406 | - | 531k |
| AQ - EIG - 10X | 0.200 | 0.284 | 0.349 | 0.379 | 0.402 | - | 522k |
| AQ - EDC - 10X | 0.186 | 0.294 | 0.326 | 0.383 | 0.408 | - | 522k |
| AQ - ERC - 10X | 0.201 | 0.292 | 0.338 | 0.383 | 0.407 | - | 522k |
| ALPF - EIG | 0.203 | 0.289 | 0.351 | 0.384 | 0.420 | - | 575k |
| ALPF - EDC | **0.220** | 0.319 | 0.363 | 0.397 | 0.420 | - | 482k |
| ALPF - ERC | 0.207 | **0.330** | **0.391** | **0.419** | **0.427** | - | 491k |
| ALPF - EIG - 10X | 0.199 | 0.293 | 0.352 | 0.391 | 0.406 | - | 581k |
| ALPF - EDC - 10X | 0.231 | 0.352 | 0.387 | 0.410 | 0.409 | - | 521k |
| ALPF - ERC - 10X | 0.235 | 0.342 | 0.382 | 0.417 | 0.426 | - | 521k |

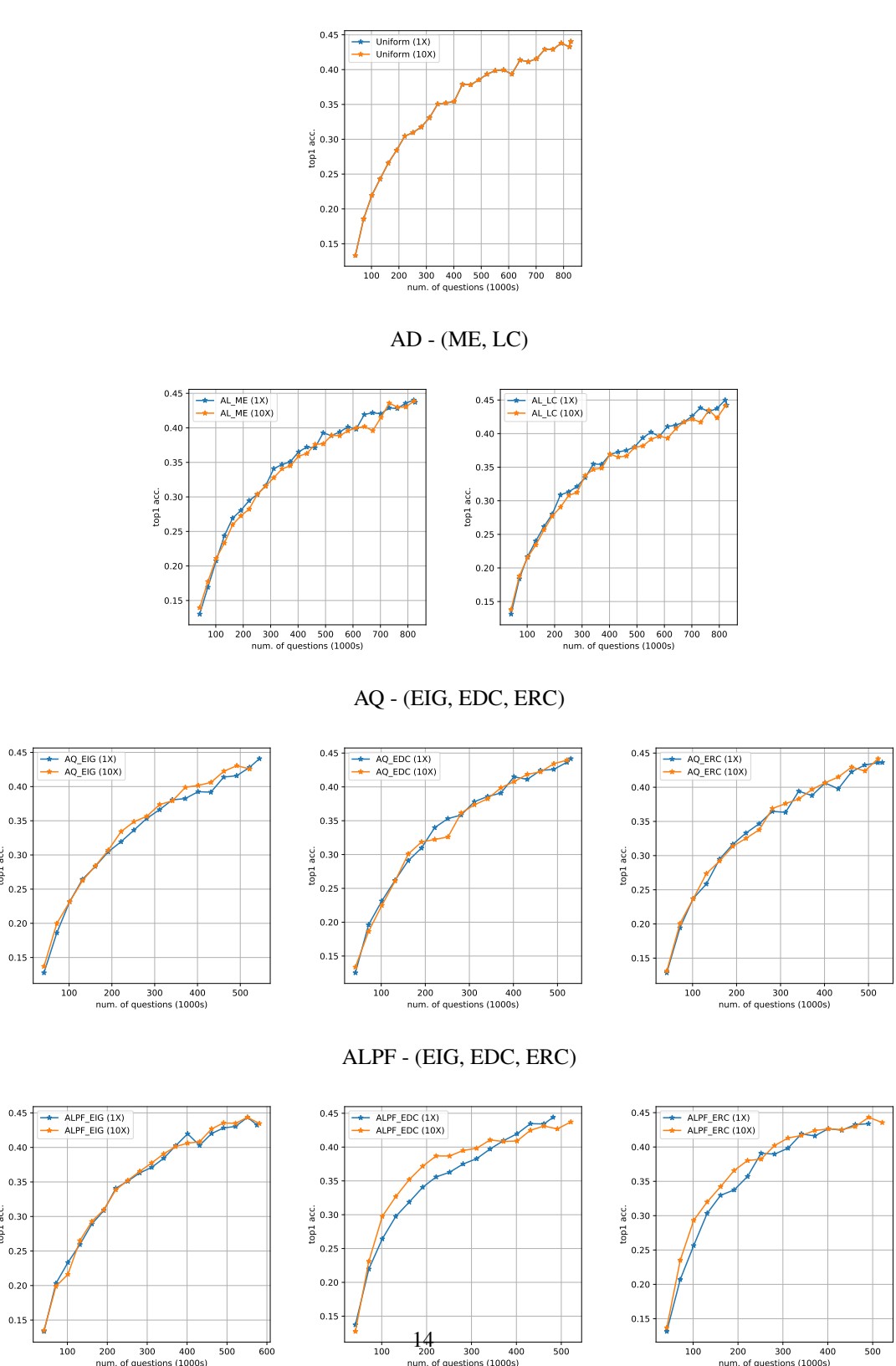

Figure 6: Updating models after every 30K questions (1X) vs. after every 3K (10X)