# OpenReview forum: "Active Learning with Partial Feedback"
_ICLR.cc/2019/Conference_

### Official Review · AnonReviewer1 · 2018-11-01
**An interesting setting combining active learning and learning with partial labesl. Nice experimental contribution, lack of conceptual insights.**

**Rating:** 7
**Confidence:** 4

**Review:**

The paper considers a multiclass classification problem in which labels are grouped in a given number M of subsets c_j, which contain all individual labels as singletons. Training takes place through an active learning setting in which all training examples x_i are initially provided without their ground truth labels y_i. The learner issues queries of the form (x_i,c_j) where c_j is one of the given subsets of labels. The annotator only replies yes/no according to whether the true label y_i of x_i belongs to c_j or not. Hence, for each training example the learner maintains a "version space" containing all labels that are consistent with the answers received so far for that example. The active learning process consists of the following steps: (1) use the current learning model to score queries (x_i,c_j); (2) query the best (x_i,c_j); (3) update the model.
In their experiments, the authors use a mini-batched version, where queries are issued and re-ranked several times before updating the model. Assuming the learner generates predictive models which map examples to probability distributions over the class labels, several uncertainty measures can be used to score queries: expected info gain, expected remaining classes, expected decrease in remaining classes. Experiments are run using the Res-18 neural network architecture over CIFAR10, CIFAR100, and Tiny ImageNet, with training sets of 50k, 50k, and 100k examples. The subsets c_j are computed using the Wordnet hierarchy on the label names resulting in 27, 261, and 304 subsets for the three datasets. The experiments show the advantage of performing adaptive queries as opposed to several baselines: random example selection with binary search over labels, active learning over the examples with binary search over the labels, and others.

This paper develops a natural learning strategy combining two known approaches: active learning and learning with partial labels. The main idea is to exploit adaptation in both choosing examples and queries. The experimental approach is sound and the results are informative. In general, a good experimental paper with a somewhat incremental conceptual contribution.

In (2) there is t+1 on the left-hand side and t on the right-hand side, as if it were an update. Is it a typo?

In 3.1, how is the standard multiclass classifier making use of the partially labeled examples during training?

How are the number of questions required to exactly label all training examples computed? Why does this number vary across the different methods?

What specific partial feedback strategies are used by AQ for labeling examples?

EDC seems to consistently outperform ERC for small annotation budgets. Any intuition why this happens?

---

> ### Author Response · Authors · 2018-11-19
> **Reply to Reviewer 1**
>
> Thanks for your feedback. We are glad that you appreciated the usefulness of the setup, the soundness of the experiments, and the insights of the results. We are also grateful for your thoughtful questions and recommendations.
>
> 1) Yes, the t+1 is a mistake. Thanks for the catch! We will fix this in the camera ready version of the paper.
>
> 2) A standard multi-class classifier cannot make use of the partially labeled data. The very purpose of these initial experiments was to establish as a sanity check that our setup for learning from partial labels with neural networks works in the first place (before adding the complexity of active learning). The point is to show that the model gets additional predictive performance as compared to if it only relied on the subset of data that had been fully annotated.
>
> 3) One key feature of ALPF is that a better algorithm identifies the correct label with a smaller number of (binary) questions. To compute the number of questions asked, we *record* the number of queries required to conclusively identify the label of every example. Note that this requires at least 1 question for each example, but may be much faster than the naive approach of drilling through the whole label hierarchy fresh for each example.
>
> Our experiments compare all three acquisition strategies with AQ (EIG, ERC, and EDC). The difference between AQ and ALPF is that AQ selects examples i.i.d., and chooses only which (possibly composite) label to query. By contrast, ALPF at each round selects both the example and the label, possibly moving on to a new example and leaving the previous example with a partial label.
>
> There are two relevant observations to the reviewer’s question. On Tiny ImageNet, ERC ends up spending the first 60K (the first two batch after warm-up) questions on just 32K distinct examples while EDC ends up querying 51K distinct examples. As we can see in Figure 2 (and not surprisingly), ERC obtains more exactly labeled examples early on, while EDC has less remaining classes overall. The fact that EDC consistently outperforms ERC early on suggests that given a very limited budget it might be better to coarsely but strategically annotate a larger dataset than to focus on obtaining more granular labels. How precisely this translates into improved classification performance is an interesting question and warrants deeper theoretical inquiry.

---

### Official Review · AnonReviewer3 · 2018-11-02
**Good paper, but lack of theoretical analysis**

**Rating:** 6
**Confidence:** 3

**Review:**

This paper proposes active learning with partial feedback, which means at each step, the learner actively chooses both which example to label and which binary question to ask, then learn the multi-class classifier with these partial labels. Three different sampling strategies are used during active learning. Experimental results demonstrate that the proposed ALPF strategy outperforms existing baselines on the predicting accuracy under a limited budget.

This paper is well-written. The main ideas and claims are clearly expressed. ALPF combines active learning with learning from partial labels. This setting is interesting and important, especially when the number of categories is large and share some hierarchical structure. The experimental results are promising. My main concern about this work is the lack of theoretical guarantees, which is usually important for active learning paper. it’s better to provide some analysis on the efficiency of ALPF to further improve the quality of the paper.
I have the following questions for the authors:
+Why vanilla active learning strategy does not work well? Which uncertainty measurement do you use here?
+The performances of this work heavily rely on the taxonomy of labels, while in some cases the taxonomy of labels is not tree structure but a graph, i.e. a label may belong to multiple hyper-labels. Can ALPF still work on these cases?

---

> ### Author Response · Authors · 2018-11-19
> **Reply to Reviewer 3**
>
> We thank the reviewer for their thoughtful feedback and were glad to see that you found our proposed setting to be both interesting and important. We would like to respond to your concerns briefly:
>
> First, concerning your questions:
> ***Re the failure of vanilla active learning***
> Since theoretical analysis guaranteeing the performance of active + deep learning has yet to be established, it’s hard to say *why* vanilla uncertainty-sampling-based active learning doesn’t work so well when applied on image classification datasets with convolutional neural networks. However, we are not the first to find this. Take for example the results Active Learning for Convolutional Neural Networks: A Core Set Approach (https://arxiv.org/pdf/1708.00489.pdf), which was published at ICLR 2018, where uncertainty sampling and even the more recent deep Bayesian active learning by disagreement perform no better than random on CIFAR 10 and only marginally better for CIFAR 100. In contrast, vanilla AL strategies have demonstrated promise on a number of NLP tasks (e.g. https://arxiv.org/pdf/1808.05697.pdf).
>
> ***Re the taxonomy of labels***
> While tree-structured taxonomies are especially convenient, our methods do not in principle depend specifically on tree structure, requiring only a list of composite labels. One can draw a parallel to general formulations of the game twenty questions where the available set of questions needn’t form tree. We thank the reviewer for the suggestion for future work and plan to evaluate our methods on with label ontologies like the MeSH labels (medical subject headings) used to annotate biomedical articles that do not form a strict tree hierarchy (some nodes have multiple parents).
>
> Regarding theoretical guarantees, we agree with the reviewer that establishing theoretical guarantees for active learning with partial labels is an especially exciting direction and plan to pursue future work in this direction. We note that generally there is a considerable gap between the theory of active learning and the practical methods established to cope with high dimensional data and modern classifiers and hope to close this gap in the future with rigorous analysis.

---

### Official Review · AnonReviewer2 · 2018-11-09
**Interesting novel Active Learning setting**

**Rating:** 7
**Confidence:** 4

**Review:**

The authors introduce a new Active Learning setting where instead of querying for a label for a particular example, the oracle offers a partial or weak label. This leads to a simpler and more natural way of retrieving this information that can be of use many applications such as image classification.

The paper is well-written and very easy to follow. The authors first present the overview of the learning scenario and then suggest three sampling strategies based on the existing AL insights (expected information gain, expected remaining classes, expected decrease in classes).

As the labels that the algorithm has to then use are partial, they make use of a standard algorithm to learn from partial labels -- namely, minimizing a partial log loss. It would be nice to properly reference related methods in the literature in Sec. 2.1.

The way of solving both the learning from partial labels and the sampling strategies are not particularly insightful. Also, there is a lack of theoretical guarantees to show value of a partial label as compared to the true label. However, as these are not the main points of the paper (introduction of a novel learning setting), I see these as minor concerns.

---

> ### Author Response · Authors · 2018-11-19
> **Reply to Reviewer 2**
>
> We thank the reviewer for their thoughtful feedback and clear recommendation to accept. We were glad to see that you found the paper to be well-articulated and easy to read.
>
> Per your feedback, we will bring up the related work (currently in section 4) and cite it throughout as each prior technical idea is introduced. Regarding the related work on partial labels are you referring to the three papers we cite later on (Grandvalet & Bengio, 2004; Nguyen & Caruana, 2008; Cour et al., 2011) or others that we missed? Please let us know if you know of other related references and we’ll be happy to add any missing citations.
>
> We agree that the choice of approaches in this paper is straightforward and meant to emphasize the importance of a novel problem setting as well as compelling experimental results. We also agree that a great next step for this work would be to establish theoretical guarantees for active learning with partial labels.

---

### Author Response · Authors · 2018-11-19
**General reply to reviewers**

We would like to thank all three reviewers for their thoughtful and detailed reviews. Overall, we were glad to see a consensus to accept the paper, with the reviews emphasizing the importance and novelty of our proposed problem setting, and the strength of our experimental work. As we continue to improve the draft, we will incorporate the constructive feedback from each reviewer. Please find replies to each review below in the respective threads.

---

### Meta-Review · Area_Chair1 · 2018-12-17
**Sets strong experimental baselines for active learning in hierarchical classification settings**

**Confidence:** 4
**Recommendation:** Accept (Poster)

**Metareview:**

This paper is on active deep learning in the setting where a label hierarchy is available for multiclass classification problems: a fairly natural and pervasive setting. The extension where the learner can ask for example labels as well as a series of questions to adequately descend the label hierarchy is  an interesting twist on active learning.  The paper is well written and develops several natural formulations which are then benchmarked on CIFAR10, CIFAR100, and Tiny ImageNet using a ResNet-18 architecture.  The empirical results are carefully analyzed and appear to set interesting new baselines for active learning.